**Particulate biogenic barium tracer of mesopelagic carbon remineralization in the**
**Mediterranean Sea (PEACETIME project)**
Stéphanie H.M. Jacquet[1*], Christian Tamburini[1], Marc Garel[1], Aurélie Dufour[1], France
VanVambeke[1], Frédéric A.C. Le Moigne[1], Nagib Bhairy[1], Sophie Guasco[1]
[1]Aix Marseille Université, CNRS/INSU, Université de Toulon, IRD, Mediterranean Institute
of Oceanography (MIO), UM 110, 13288 Marseille, France
[*]Correspondence to: S. Jacquet (*stephanie.jacquet@mio.osupytheas.fr*)
**PEACETIME special issue**

**ABSTRACT**

We report on the sub-basins variability of particulate organic carbon (POC) remineralization in the western and central Mediterranean Sea in late spring during the PEACETIME cruise. POC remineralization rates were estimated using the excess biogenic particulate barium ($Ba_{xs}$) inventories in the mesopelagic layers (100-1000 m depth) and compared with prokaryotic heterotrophic production (PHP). $Ba_{xs}$-based mesopelagic remineralization rates (MR) ranged from $25 \pm 2$ to $306 \pm 70$ mg C m$^{-2}$ d$^{-1}$. MR were larger in the Algero-Provençal (ALG) basin than in the Tyrrhenian (TYR) and Ionian (ION) basins. Our $Ba_{xs}$ inventories and integrated PHP data also indicated that significant mesopelagic remineralization occurred down to 1000 m depth in the ALG basin in contrast to the ION and TYR basins, where remineralization was mainly located above 500 m depth. We proposed that the higher and deeper MR rates in the ALG basin were sustained by an additional particles export event driven by deep convection. The TYR basin (in contrast to the ALG and ION basins) presented the impact of a previous dust event as reflected by our particulate Al water column concentrations. The ION and TYR basins showed small-scale heterogeneity in remineralization processes, reflected by our $Ba_{xs}$ inventories and integrated PHP data at the #Tyrr long duration station. This heterogeneity was linked to the mosaic of blooming and non-blooming patches reported in this area during the cruise. In contrast to the western Mediterranean Sea (ALG basin), the central Mediterranean Sea (ION and TYR basins) showed lower remineralization rates restricted to the upper mesopelagic layer during the late spring PEACETIME cruise.

## 1. Introduction

In the ocean, remineralization rate associated with sinking particles is a crucial variable for air sea $CO_2$ balance [Kwon et al., 2009]. Most of the sinking particulate organic carbon (POC) conversion (i.e. remineralization) into $CO_2$ by heterotrophic organisms (i.e. respiration) occurs within the mesopelagic zone (100-1000 m) [Martin et al., 1987; Buesseler et al., 2007; Buesseler and Boyd, 2009]. A quantitative representation of this process is thus crucial to future predictions of the ocean's role in the global C cycle [IPCC, 2014]. Particulate biogenic barium ($Ba_{xs}$) is a geochemical tracer of POC remineralization in the mesopelagic layer. $Ba_{xs}$ occurs in the form of barite ($BaSO_4$ crystals) in the dark ocean as a byproduct of prokaryotic remineralization. In a global ocean undersaturated with respect to barite [Monnin and Cividini, 2006], $Ba_{xs}$ precipitates inside oversaturated biogenic micro-environments during POC degradation by heterotrophic prokaryotes, through sulfate and/or barium enrichment [Dehairs et al., 1980; Stroobant et al., 1991; Bertram and Cowen, 1997; Ganeshram et al., 2003]. By applying a transfer function relating $Ba_{xs}$ to $O_2$ consumption [Dehairs et al., 1997] $Ba_{xs}$ has been widely used since the 90's as an estimator of mesopelagic POC remineralization rates in various sectors of the Southern Ocean, North Pacific and North Atlantic [Cardinal et al., 2001, 2005; Dehairs et al., 2008; Jacquet et al., 2008a, 2008b, 2011, 2015; Planchon et al., 2013; Lemaitre et al., 2018]. Jacquet et al. [2021] recently reported that such transfer function could be applied in the Mediterranean Sea without restriction. This last study complemented previous investigations aiming at improving the use of $Ba_{xs}$ to estimate local processes of POC remineralization in the Mediterranean Sea [Jacquet et al., 2016; Jullion et al., 2017]. The Mediterranean Sea represents a unique case study, mainly due to unresolved issues related to the imbalance in the regional C budget such as the coupling between surface biology and deeper remineralization, timescales of their variability between basins and discrepancies between mesopelagic trophic structure and respiration dynamics

[Sternberg et al., 2007, 2008; Santinelli et al., 2010; Lopez-Sandoval et al., 2011; Luna et al.,
2012; Tanhua et al., 2013b; Malanotte-Rissoli et al., 2014].

65         The present work is part of the PEACETIME project (ProcEss studies at the Air-sEa

Interface after dust deposition in the MEditerranean sea) (http://peacetime-project.org/). It
aimed at studying the impact of atmospheric Saharan dust on the Mediterranean
biogeochemistry [Guieu et al., 2020a]. Dust deposition is a major source of macro and micro
nutrients and ballasted material to surface waters that likely impacts the biological carbon
pump through organic matter production (i.e. primary production) and its subsequent export
and remineralization in the water column [Pabortsava et al., 2017; Gazeau et al., 2021].
Overall, the aims of the present contribution to the PEACTIME project were: (1) to document
particulate biogenic $Ba_{xs}$ in different ecoregions of the western and central parts of the
Mediterranean Sea. Previous $Ba_{xs}$ data in the Mediterranean Sea are relatively scarce with
limited vertical sampling resolution [Sanchez-Vidal et al., 2005] or restricted locations
[Dehairs et al., 1987; Sternberg et al., 2007, 2008; van Beek et al., 2009]; (2) to determine the
relationship between $Ba_{xs}$ and environmental variability, including dust deposition, (3) to
estimate $Ba_{xs}$-based POC remineralization rates (MR) at mesopelagic depths using the
Dehairs' transfer function [Dehairs et al., 1997] which we have recently validated for the
Mediterranean Sea [Jacquet et al., 2021], and (4) assess potential differences in
remineralization length scale of POC in the various ecoregions of the Mediterranean Sea.

**2. Material and methods**
**2.1 Study area**

85       The PEACETIME cruise (https://doi.org/10.17600/17000300) was conducted during late

spring from May 10 to June 11, 2017 (French R/V Pourquoi pas?) in the western and central
Mediterranean (Figure 1a). The Mediterranean Sea is a semi-landlocked sea, with limited but
crucial exchange with the Atlantic Ocean, two deep overturning cells, one shallow circulation
and a complex upper layer circulation with several permanent and quasi-permanent eddies.
The hydrography during the PEACETIME cruise was characterized by three-layers:
surface, intermediate and deep waters, typical for the Mediterranean [Tamburini et al., 2013;
Tanhua et al., 2013a; Hainbucher et al., 2014, Malanotte-Rizzoli et al., 2014]. Briefly, the
main water masses are (see potential temperature – salinity diagram in Figure 1b): (1) from
west to east surface Atlantic Water (SW) is gradually replaced by Ionian surface Water (ISW)
and Levantine Surface water (LSW); (2) Winter Intermediate Water (WIW) and Levantine
Intermediate water (LIW). LIW is present at intermediate depths (from 200 to 800 m) and is
characterized by a local maximum of salinity and a local minimum of dissolved oxygen
concentration; (4) Mediterranean Deep Water (MDW).
Three main ecoregions [Reygondeau et al., 2017; Ayata et al., 2018] were crossed during
the cruise: the Algero-Provençal basin (later referred to as ALG), the Tyrrhenian basin (TYR)
and the Ionian basin (ION) (Figure 1a). These basins displayed the typical eastward
oligotrophic gradient as reported in previous studies [Moutin and Raimbault, 2002; Durrieu
de madron et al., 2011; Pujo-Pay et al., 2011; Tanhua et al., 2013a; Reygondeau et al., 2017;
Guieu et al., 2020a]. However, this trend was not homogeneous, as for instance in the Ionian
Sea (a crossroad of waters of contrasted biological history) where a mosaic of blooming and
non-blooming areas co-occured in spring [Berline et al., 2021]. A diatom-dominated deep
chlorophyll maximum that coincided with a maximum in biomass and primary production
(PP) was well developed and observed all along the cruise track (Marañón et al., 2021). PP is
described in details in Van Wambeke et al. (this issue). Furthermore, important dust
deposition affected the TYR basin a few days before our arrival at stations #Tyrr and #5,
while in the ALG basin, dust deposition occurred few hours before our sampling at station
#Fast (Bressac et al., this issue). POC downward fluxes measured at 200 m depth were similar
at the 3 long stations (#Fast, #Tyrr and #Ion).

**2.2 Barium sampling and sample processing**

Thirteen stations were sampled for particulate barium from the surface to 2000 m (thirty
depths in total) in the ALG (stations #1, #2, #3, #10, #Fast, #9 and #4), TYR (stations #5,
Tyrr and #6) and ION (stations #8, #7 and #Ion) basins (Table 1). Three of these stations were
sampled twice on different days (long duration stations #Fast, #Tyrr and #Ion), but due to
technical problem no particulate barium data are available for the second visit at station #Ion.
Three days separate both visits at station #Fast and two days at #Tyrr.
For particulate barium, 4 to 6 L of seawater sampled using Niskin bottles were filtered onto
47 mm polycarbonate membranes (0.4 μm porosity) under slight overpressure supplied by
filtered air. Filters were rinsed with a few mL of MQ grade water to remove sea salt, dried
(50°C) and stored in Petri dishes for later analysis. In the laboratory, we performed a total
digestion of filters using a concentrated tri-acid (0.5 mL HF /1.5 mL $HNO_3$ / HCl 1 mL; all
Optima grade) mixture in closed teflon beakers overnight at 95°C in a clean pressurized room.
After evaporation close to dryness, samples were re-dissolved into 10 mL of 2% $HNO_3$.
Subsequently, samples were analysed for Ba and other elements of interest (i.e. Al, Na, Sr and
Ca) by HR-ICP-MS (High Resolution-Inductively Coupled Plasma- Mass Spectrometry;
ELEMENT XR, Thermo). Based on analyses of external certified reference standards,
accuracy and reproducibility were both within ±5%. More details on sample processing and
analysis are given in Cardinal et al. [2001] and Jacquet et al. [2015]. The presence of sea-salt
was checked by analysing Na and the sea-salt particulate Ba contribution was found to be
negligible (<0.1% of total Ba). Particulate biogenic barium in excess (hereafter referred to as
$Ba_{xs}$) was calculated as the difference between total Ba and lithogenic Ba. The lithogenic Ba
concentration was determined using Al concentration and the upper continental crust (UCC)
Ba:Al molar ratio [Dymond et al., 1992; Taylor and Mc Lennan, 1985]. The standard
uncertainty [Ellison et al., 2000] on $Ba_{xs}$ concentration ranges between 5.0 and 5.5%. The
term "in excess" is used to indicate that concentrations are larger than the $Ba_{xs}$ background
(Ba BKG). The background (or residual value) is considered as "preformed" $Ba_{xs}$ at zero
oxygen consumption left over after transfer and partial dissolution of $Ba_{xs}$ produced during
degradation of previous particles export events. This background $Ba_{xs}$ value likely depends on
the saturation state of the water with respect to barite ($BaSO_4$, the main phase of particulate
biogenic barium). Saturation indexes were reported in Jacquet et al. [2016] over a high
resolution and quasi-zonal Mediterranean transect (M84/3 cruise; Tanhua et al., 2013a,
2013b). They revealed that the water column throughout the study area is largely
undersaturated, with saturation state ranging between 0.2 and 0.6. A background $Ba_{xs}$ value of
130 pM was recently reported in Jacquet et al. [2021]. It is close to the average $Ba_{xs}$ contents
observed at greater depth (>1000 m) in the present study (see below).

**2.3 Prokaryotic heterotrophic production**
Prokaryotic heterotrophic production (PHP) estimation was measured by the L-[4,5-3H]-
Leucine ($^3$H-Leu, specific activity 109 Ci mmol$^{-1}$, PerkinElmer®) incorporation technique
(Kirchman, 1993). Details of the protocols can be found in Van Wambeke et al. (2021).
Briefly, in epipelagic layers (0-200 m) 1.5 ml seawater samples were incubated at 20 nM $^3$H-
Leu final concentration using the microcentrifuge technique (Smith and Azam, 1992). For the
mesopelagic layers, 20 ml (200-800 m depth) and 40 ml (below 800 m depth) seawater
samples were incubated using 20 nM and 10 nM $^3$H-Leu final concentration, respectively,
using the filtration technique (Tamburini et al., 2002). Samples were incubated at in situ
temperature. To calculate PHP, we used the empirical conversion factor of 1.5 ng C per pmol

162 of incorporated leucine assuming that isotopic dilution was negligible under saturating

163 concentrations of leucine as checked occasionally from concentration kinetics (Van Wambeke

164 et al., 2021).


166 **2.4 POC remineralization rates**

167 We recently reported on the validity of the Dehairs's transfer function [Dehairs et al., 1997]

168 in the Mediterranean basin to estimate mesopelagic POC remineralization [Jacquet et al.,

169 2021]. We applied the similar approach to estimate remineralization rates (MR):

170 $MR = [(Ba_{xs} - Ba\ BKG)/17450] \times Z \times RR$ (Eq.1)

171 where $Ba_{xs}$ is the depth-weighted average $Ba_{xs}$ concentration (DWA; pM), i.e. the $Ba_{xs}$

172 inventory divided by the depth layer considered Z, and RR the Redfield $C/O_2$ molar ratio

173 (127/175; Broecker et al., 1985). As reported above, a Ba BKG concentration of 130 pM was

174 used. MR rates were then integrated over the 100-500 m (upper mesopelagic zone) and 100-

175 1000 m (entire mesopelagic zone) depth layers.

176

177 **3. Results**

178 Particulate biogenic $Ba_{xs}$, biogenic Ba fraction (%) and particulate Al, Sr and Ca

179 concentrations are reported in Figure 2 in the upper 2000 m of the water column along a zonal

180 transect crossing the three main sub-basins. PHP rates are also reported in Figure 2 in the

181 upper 500 m along the same transect.

182 $Ba_{xs}$ displayed a similar water column distribution as reported in various sectors of the

183 global Ocean, i.e. relatively low surface concentrations, a maximum in the mesopelagic layer

184 (100-1000 m) followed by a decrease of concentrations back to a background level at deeper

185 depths, usually 1000 m (Figure 3). At stations #9, #Tyrr, #8 and #Ion, $Ba_{xs}$ concentrations in

186 the upper 100 m were quite high (>5000 pM), with values reaching up to 11700 pM at 80 m

depth at station #Tyrr (Figure 2a). Such high $Ba_{xs}$ concentrations in the upper layers are quite
unusual, though similar values (up to 9000 pM) were occasionally observed in earlier
Southern Ocean studies [Dehairs et al., 1991, 1997; Jacquet et al., 2007b, 2008a, 2008b]. The
high $Ba_{xs}$ values at stations #Tyrr and #Ion were associated with higher Sr (up to 4267 pM at
100 m at station #Ion) and Ca (>130 nM) concentrations (Figure 2d, e). Thoughtout the water
column, Sr and Ca concentrations ranged from to 448 to 6938 pM and from 30 to 488 nM,
respectively. Sr/Ca molar ratios ranged from 7 to 45 mmol mol$^{-1}$whithin the range of ratios
reported in organic material [Martin and Knauer, 1976]. The upper mesopelagic layer (100-
500 m) showed the characteristic Ba excess (maximum), as illustrated in Figure 3a. The
lithogenic impact on the $Ba_{xs}$ signal was relatively low (<20 %) except at stations #4, #5 and
#Tyrr where it reached up to 30 % at some depths in the water column (Figures 2b, 3b). High
$Ba_{xs}$ concentrations at stations in the ALG basin and at station #7 in the ION basin extended
deeper than at stations in the TYR basin (Figure 2a). At station #Ion $Ba_{xs}$ maximum coincided
with higher Ca (up to 186 nM) concentrations in the upper mesopelagic layer (Figure 2e).
However, the $Ba_{xs}$ maximum also extented deeper. This is especially salient at stations in the
ALG basin. At the other stations $Ba_{xs}$ concentrations below 500 m decreased to reach the
background value of around 130 pM. Among stations sampled twice for barium during the
cruise, station #Fast (ALG basin) presented similar $Ba_{xs}$ profiles except between 400 and
1000 m depth where lower concentrations were measured during the second visit (3 days
later; Figure 3a). The $Ba_{xs}$ signal was mostly biogenic and rather stable over the whole water
column at this station. This was also the case at station #Ion. In contrast, at station #Tyrr
differences between $Ba_{xs}$ profiles mainly occurred in the surface layer and upper mesopelagic
layer, with relatively higher $Ba_{xs}$ peaks during the second visit (2 days later; Figure 3b). The
biogenic Ba fraction was also more variable throughout the water column at #Tyrr.

211        PHP rates decreased from west to east in surface waters (Figure 2f). At station #Fast,

PHP rates decreased from 49 ng C $l^{-1}$ $h^{-1}$ in surface to values between 7 and 11 ng C $l^{-1}$ $h^{-1}$ at
100 m depth and below 6 ng C $l^{-1}$ $h^{-1}$ below 200 m depth (Figure 3d). Same trends were found
at #Tyrr and #Ion with values in surface waters around 36 and 25 ng C $l^{-1}$ $h^{-1}$ respectively
(Figures 3e and 3f).

216        Depth-weighted average (DWA) concentrations of $Ba_{xs}$ are reported in Table 1 and

Figure 5 for the upper (100-500 m) and entire (100-1000 m) mesopelagic layer. Since the base
of the mixed layer was shallower than 100 m, this depth is taken as the upper boundary of the
mesopelagic domain. DWA values ranged from 221 to 979 pM. On average, stations located
in the ALG basin presented higher DWA than in the TYR and ION basins. DWA $Ba_{xs}$ values
remained rather stable over the 3–day period at station #Fast between 100 and 500 m depth,
but decreased in deeper layers (Figure 5). As a consequence, the DWA changed from 527 to
381 pM for the entire 100-1000 depth layer. In contrast, at station #Tyrr DWA $Ba_{xs}$ values for
the 100-500 m and 100-1000 m depth layers increased over the 2–day period (from 284 to
542 pM and from 200 to 380 pM, respectively). On average DWA $Ba_{xs}$ reached 577±286,
378±123 and 529±213 pM (100-500 m), and 527±288, 280±82 and 358±112 pM (100-1000
m) in the ALG, TYR and ION basin, respectively.

**4. Discussion**
**4.1 $Ba_{xs}$ distributions across the sub-basins**

231        The very high $Ba_{xs}$ concentrations reported in the surface layer at stations #9, #Tyrr,

#8 and #Ion were associated with local Sr and Ca maxima, likely linked to potentially
ballasted phytoplankton-derived material. Similar observations were previously reported in
the Southern Ocean, revealing that in the surface water particulate $Ba_{xs}$ is incorporated into or
adsorbed onto biogenic material, with barite being a minor component [Dehairs et al., 1991,
1997; Jacquet et al., 2007a, 2008a, 2008b]. In deeper layers, $Ba_{xs}$ presented the characteristic

maximum reflecting mesopelagic remineralization processes. Mesopelagic $Ba_{xs}$ distributions presented here were similar to those reported in Jacquet et al. [2021] and Sternberg et al. [2008] in the northwestern Mediterranean Sea (ANTARES and DYFAMED observatory sites, respectively). The $Ba_{xs}$ maximum extended down to 1000 m depth in the ALG basin, while it was mostly located in the upper 500 m depth in the TYR basin. The lithogenic impact on the $Ba_{xs}$ signal was relatively very low (<5%), except at stations #4, #5 and #Tyr where it was more variable and reached up to 30 % at some depths (Figure 2b and 2c). A large dust deposition event occurred over a large area including the southern Tyrrhenian Sea just before the beginning of the PEACETIME campaign. Particulate Al concentrations and estimated lithogenic Ba fraction were sampled at these stations 5 to 12 days after the event and reflected the impact of this dust event in depth. These conclusions are further supported by results reported in Bressac et al. [this issue], showing that Saharan dust depositions strongly impacted Stations #4, #5, #Tyr and #6 whereby a significant fraction of dust particles was transferred to mesopelagic depths.

## 4.2 Mesopelagic $Ba_{xs}$ and prokaryotic heterotrophic production

Previous studies highlighted the relationship between the mesopelagic $Ba_{xs}$ and the vertical distribution of prokaryotic heterotrophic production (PHP), reflecting the temporal progression of POC remineralization processes. In mesopelagic layers, $Ba_{xs}$ content is smaller when most of the PHP occurs in the upper mixed layer (indicating an efficient, close to complete remineralization within the surface), compared to situations where a significant part of PHP is located deeper in the water column (reflecting significant deep prokaryotic activity and POC export). Figure 3 shows the PHP profiles at long station #Fast, #Tyr and #Ion (see also VanWambeke et al. [2021] for more detail on PHP). Figure 4 shows the ratio of integrated surface (100 m) to integrated upper mesopelagic (500 m) PHP vs. DWA $Ba_{xs}$

calculated over the 100-500 m depth interval. Results are confronted to the data obtained in
the Southern Ocean [Jacquet et al., 2008a, 20015] and recently in the northeast Atlantic and
northwestern Mediterranean Sea (PAP and ANTARES observatory sites, respectively)
[Jacquet et al., 2021]. The blue line in Figure 4 represents the trend obtained during KEOPS2
[Jacquet et al., 2015]; it does not include encircled data points referred to as "season
advancement". Results during PEACETIME followed a similar trend than found for KEOPS2
with higher DWA $Ba_{xs}$ in situation where a significant part of column-integrated PHP is
located deeper in the water column (high Int.PHP100/Int.PHP500 ratio, Figure 4). Note that
some data points, characterized by low DWA $Ba_{xs}$ values, did not follow the trend from
KEOPS2 (stations #3, #5 and #Tyrr2). During KEOPS2, the lowest DWA were reported for
stations located in a meander and reflecting different (earlier) stages of a bloom compared to
the other stations (see "season advancement" in Figure 4). Similarly, station #5 and #Tyrr2
reflected the temporal evolution of the establishment (or advanced stages) of mesopelagic
remineralization processes in the TYR basin compared to the other basins. Measurements
carried out during the second visit at station #Tyrr two days later corroborated this hypothesis
showing an increase in remineralization in the upper mesopelagic layer (DWV $Ba_{xs}$ increased
from 284 to 542 pM). At the DYFAMED station, Sternberg et al. [2008] reported the seasonal
evolution of $Ba_{xs}$ profiles on a monthly basis between February and June 2003. These authors
showed the mesopelagic $Ba_{xs}$ build up (and barite stock increase) following the spring
phytoplankton bloom development, enhanced POC fluxes and subsequent remineralization.
Overall, DWA $Ba_{xs}$ reported in the present study were higher than those reported by
Sternberg et al. [2008] (maximum of 463 pM; 0-600 m). The variability over the two days
period at station #Tyrr was of the same order of magnitude as the seasonal DWA $Ba_{xs}$
dynamics found at DYFAMED and similar to changes found over a few days to week period
in different sectors of the Southern Ocean [Cardinal et al., 2005; Jacquet et al., 2007a; 2015].
The column-integrated PHP vs. DWA $Ba_{xs}$ ratio at station #Tyrr confirms that the second
occupation experienced higher remineralization rates in the upper mesopelagic layer than
during the first one (Table 1).

**4.3 Mesopelagic C remineralization**
POC remineralization rates (MR) estimated from DWA $Ba_{xs}$ values using Eq. (1) are shown
in Figure 5 for the upper (100-500 m) and entire (100-1000 m) mesopelagic layer together
with primary productivity [van Wambecke et al., 2021]. MR ranged from $25 \pm 2$ to $306 \pm 70$
mg C $m^{-2}$ $d^{-1}$ and primary production ranged from 138 to 284 mg C $m^{-2}$ $d^{-1}$. Large difference
in MR between the upper and the whole mesopelagic layers can be seen in the ALG basin.
This is more pronounced at station #9 with MR of 91 mg C $m^{-2}$ $d^{-1}$ in the upper (100 to 500 m
depth) layer and 306 mg C $m^{-2}$ $d^{-1}$ in the entire mesopelagic layer (Figure 5). These results
show that significant remineralization occurred between 500 and 1000 m in the ALG basin in
contrast to the ION and TYR basins where remineralization occurred mainly in the
mesopelagic layer between 100 and 500 m depth. Similar conclusion was reached by Jullion
et al. [2017] from dissolved Ba and Parametric Optimum Multiparameter (POMP)-derived
POC remineralization rates along a zonal transect between the Lebanon coast and Gibraltar
(from 156 to 348 mg C $m^{-2}$ $d^{-1}$; M84/3 cruise, April 2011). Independent of any dust input
considerations, Jullion et al. [2017] showed significant differences in the mesopelagic MR
between the western and eastern Mediterranean, indicating an additional organic carbon
export pathway to depth. The western basin is indeed the site of deep shelf and open ocean
convection, transferring organic matter to deeper layers [Durrieu de Madron et al., 2013;
Stabholz et al., 2013]. The larger MR fluxes found in the ALG basin during PEACETIME are
in line with an ecoregion with recurrent injection of material by winter convection (hypothesis
of particle injection pump; Boyd et al. [2019]), sustaining higher rates of remineralization
below 500 m depth. In contrast in the TYR basin remineralization was mainly located in the
upper mesopelagic layer. Stations in the TYR basin received dust inputs a few days before our
arrival at these stations; the particulate Al concentrations and estimated lithogenic Ba fraction
reflected the impact of this event (Figure 2; Bressac et al., this issue). At station #Tyrr the
DAW $Ba_{xs}$ vs. column-integrated PHP increase between the two visits indicated higher MR
rates. MR were mainly localized in the upper 500 m. Another atmospheric deposition event
occurred on June 5, a few hours after the first sampling at station #Fast in the ALG basin.
However, station #Fast does not present any evidence of an impact at mesopelagic depths on
particulate Al concentrations and estimated lithogenic Ba. In contrast to conditions in the
surface mixed layer, the generation of an observable signal from the mesopelagic
remineralization and subsequent $Ba_{xs}$ formation to a single dust event would require more
time than the time span between atmospheric deposition and sampling at #Fast (in contrast to
station #Tyr where the dust event occurred 5 to 12 days before). In the ION basin where
stations did not reflect the impact of any deposition event and were not subject to potential
deep convection, DWA $Ba_{xs}$ and MR fluxes were mostly restricted to the upper mesopelagic
layer. Berline et al. [this issue] report small-scale heterogeneity of particles abundances at
ION stations, emphasizing the spatial decoupling between particle production and particle
distribution and adding complexity in estimating the time lag between production and export
of particles, and thus C transfer eat depth [Stange et al., 2017; Henson et al., 2011]. Further,
no significant surface production event occurred in the ION basin However, surface particles
at station #8 seemed related to a past production event without significant vertical export by
the time the station was sampled. As reported in Van Wambeke et al. [this issue], primary
production fluxes were slightly higher in the ION basin (from 158 to 208 mg C $m^{-2}$ $d^{-1}$) than
in the TYR basin (from 142 to 170 mg C $m^{-2}$ $d^{-1}$). Overall, DWA $Ba_{xs}$ and MR fluxes reported
in the ION basin would thus reflect earlier stage of export and remineralization processes. The
same applies to station #Tyrr 2 (in contrast to #Tyrr4) according to the DWA $Ba_{xs}$ vs
integrated-PHP trend.

**5. Conclusion**

The present paper expands the data coverage of $Ba_{xs}$ distribution in the ALG, TYR and
ION basins (western and central Mediterranean Sea) in late spring 2017. Results highlight that
mesopelagic remineralization processes are mainly located in the upper 500 m horizon in the
TYR and ION basins, while they occur in the lower mesopelagic zone (down to 1000 m) in
the ALG basin. We suggest that particle injection driven by the seasonal winter deep
convection in the western basin would sustain the larger and deeper MR rates we observed in
the ALG basin. At both TYR and ION basins, $Ba_{xs}$ indicated lower (intensity) and upper
mesopelagic-layer restricted remineralization processes that could be the results of a previous
dust deposition event (in particular at #Tyrr) or the patchiness of time lags between
production and export of particles.

**Data availability**

Guieu et al., Biogeochemical dataset collected during the PEACETIME cruise. SEANOE.
https://doi.org/10.17882/75747 (2020).

**Author contributions**

SJ wrote the manuscript with the contribution of all co-authors. SJ and A. Dufour managed
barium analyses; CT, MG, SG and FVV managed PHP analyses. MG, SG and NB performed
Ba sampling during the cruise.

**Competing interests**

The authors declare that they have no known competing financial interests or personal
relationships that could have appeared to influence the work reported in this paper.

**Special issue statement**

This article is part of the special issue "Atmospheric deposition in the low-nutrient–low-chlorophyll (LNLC) ocean: effects on marine life today and in the future (ACP/BG interjournal SI)". It is not associated with a conference.

369

**Financial Support**

The project leading to this publication received funding from CNRS-INSU, IFREMER, CEA, and Météo-France as part of the program MISTRALS coordinated by INSU (doi: 10.17600/17000300). The instrument (ELEMENT XR, ThermoFisher) was supported in 2012 by European Regional Development Fund (ERDF).

375

**Acknowledgments**

This study is a contribution to the PEACETIME project (http://peacetime-project.org), a joint initiative of the MERMEX and ChArMEx components. PEACETIME was endorsed as a process study by GEOTRACES. It is also a contribution to SOLAS and IMBER. We thank the captain and the crew of the RV Pourquoi Pas? for their professionalism and their work at sea. We warmly thank C. Guieu and K. Deboeufs, as coordinators of the program PEACETIME and chiefs scientists of the campaign. This work is a contribution to the "AT – POMPE BIOLOGIQUE" of the Mediterranean Institute of Oceanography (MIO).

**Figures**

Figure 1: (a) Map of the study area showing the three sub-basins (ALG, TYRR and ION) with stations' locations. The dashed line represents the zonal transect reported in Figure 2; (b) Potential temperature – salinity diagram with isopynals (kg m$^{-3}$) for PEACETIME profiles. Graph produced using Ocean Data View (Schlitzer, 2002).

Figure 2: Sections of (a) particulate biogenic Ba (Ba$_{xs}$, pM), (c) Al (pM), (d) Sr (pM) and (e) Ca (nM) concentrations, and (b) % biogenic Ba (Ba$_{xs}$) in the upper 2000 m water column. (f) Section of PHP (ngC L$^{-1}$ h$^{-1}$) in the upper 500 m of the water column. Graph produced using Ocean Data View (Schlitzer, 2002).

Figure 3: (a-c) Ba$_{xs}$ (pM) and (d-f) (ngC L$^{-1}$ h$^{-1}$) profiles in the upper 2000 m and 1000 m of the water column, respectively, at long stations #Fast, #Tyrr and #Ion. (a-c) The dashed grey line represents the Ba$_{xs}$ background (BKG) and the grey area represents the fraction for which Ba$_{xs}$ is mostly biogenic.

Figure 4: Ratio of surface layer integrated PHP (Int.PHPx1) to mesopelagic integrated PHP (Int.PHPx2) versus mesopelagic depth-weighted average (DWA) Ba$_{xs}$ (pM) during PEACETIME. The same data are reported for the KEOPS1 and KEOPS2 cruises (Southern Ocean; Jacquet et al., 2015) and at the PAP (NE-Atlantic) and ANTARES/EMSO-LO (NW-Mediterranean Sea) observatory sites (Jacquet et al., 2021). The blue line (R$^2$=0.88) represents the trend reported during KEOPS2 (Jacquet et al., 2005). The data points referred to as "season advancement" (encircle by the blue line) were excluded from the KEOPS2 regression analysis shown here.


Figure 5: integrated POC remineralization rates (mg C m$^{-2}$ d$^{-1}$) in the upper (100 to 500 m
depth) and entire (100 to1000 m depth) mesopelagic layer in the ALG, TYR and ION basins.
Open squares represent primary production (mg C m$^{-2}$ d$^{-1}$; Van Wambeke et al., this issue).


**Tables**
Table 1: Depth-weighted average (DWA) concentrations of $Ba_{xs}$ (pM) and remineralization
rates (MR; mg C m$^{-2}$ d$^{-1}$) for the upper (100 to 500 m depth) and entire (100 to 1000 m depth)
mesopelagic layer.

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

Figure 1

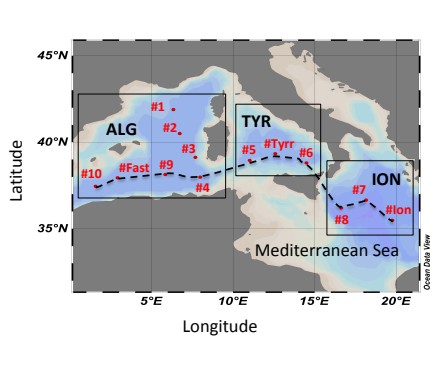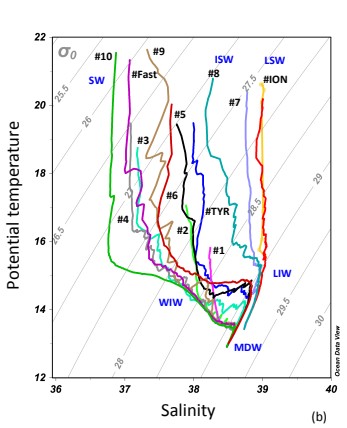






Figure 2

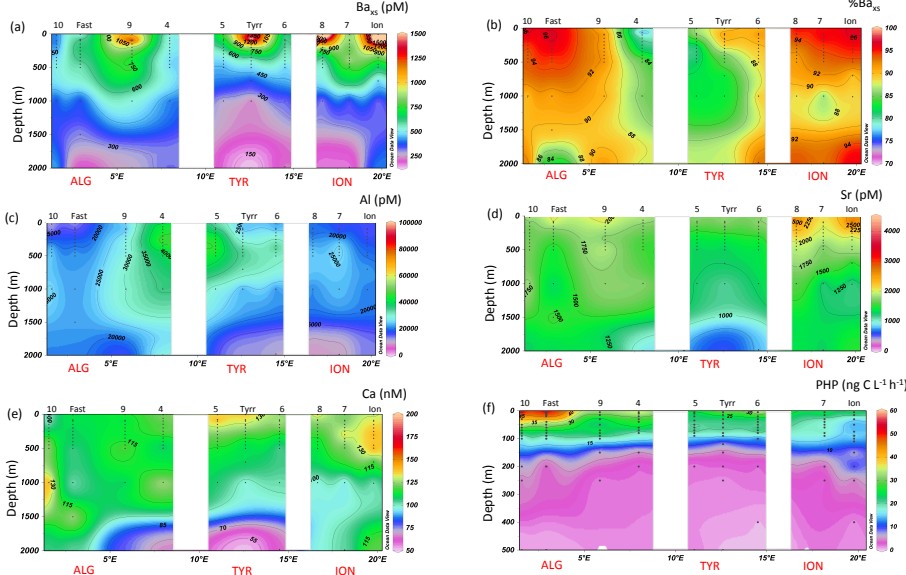




Figure 3

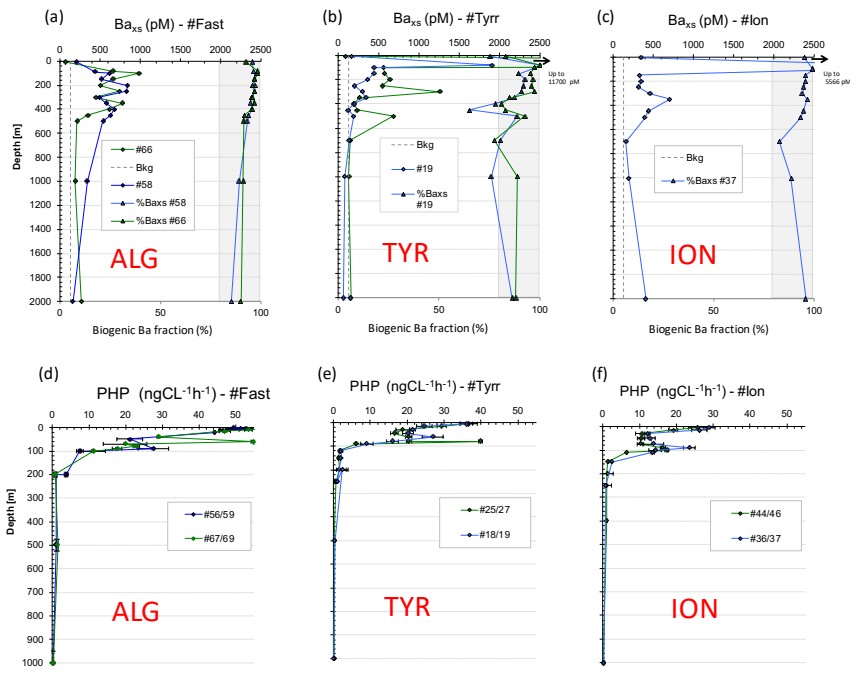




Figure 4

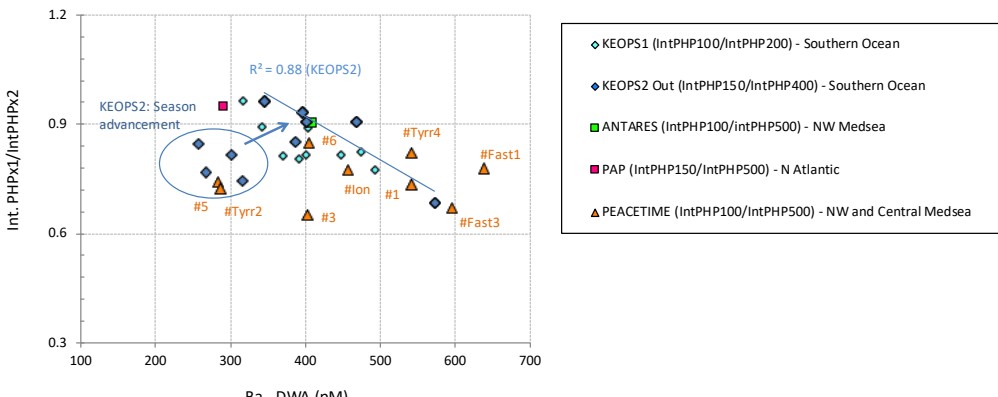





Figure 5

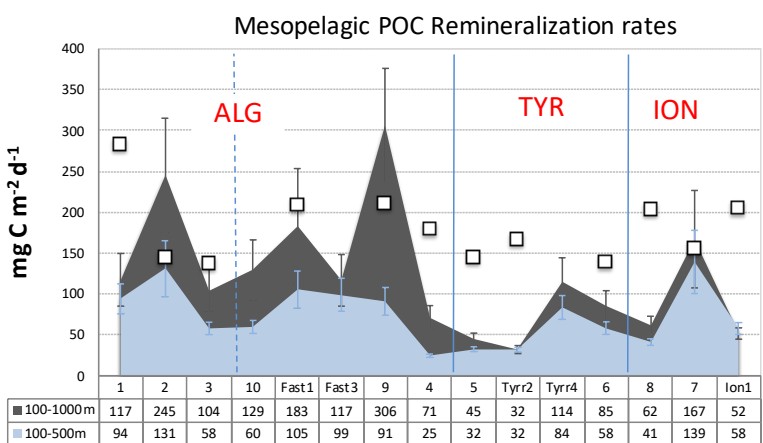




Table 1

| Basin | Station # | Mesopelagic layer | DWA Ba$_{xs}$ [pM] | MR [mg C m$^{-2}$ d$^{-1}$] | MR Stnd error [%] |
|---|---|---|---|---|---|
| Algero-Provençal | 1 | upper | 542 | 94 | 20 |
| | 1 | entire | 374 | 117 | 27 |
| | 2 | upper | 717 | 131 | 26 |
| | 2 | entire | 645 | 245 | 28 |
| | 3 | upper | 402 | 58 | 13 |
| | 3 | entire | 353 | 104 | 25 |
| | 4 | upper | 243 | 25 | 8 |
| | 4 | entire | 281 | 71 | 20 |
| | 9 | upper | 981 | 91 | 19 |
| | 9 | entire | 979 | 306 | 23 |
| | Fast1 | upper | 638 | 105 | 21 |
| | Fast1 | entire | 527 | 183 | 38 |
| | Fast3 | upper | 596 | 99 | 20 |
| | Fast3 | entire | 381 | 117 | 27 |
| | 10 | upper | 418 | 60 | 13 |
| | 10 | entire | 410 | 129 | 29 |
| Tyrrhenian | 5 | upper | 283 | 32 | 9 |
| | 5 | entire | 226 | 45 | 17 |
| | Tyrr2 | upper | 284 | 32 | 9 |
| | Tyrr2 | entire | 200 | 32 | 15 |
| | Tyrr4 | upper | 542 | 84 | 17 |
| | Tyrr4 | entire | 380 | 114 | 26 |
| | 6 | upper | 404 | 58 | 13 |
| | 6 | entire | 313 | 85 | 22 |
| Ionian | 7 | upper | 769 | 139 | 28 |
| | 7 | entire | 485 | 167 | 36 |
| | ION | upper | 456 | 58 | 13 |
| | ION | entire | 315 | 52 | 13 |
| | 8 | upper | 363 | 41 | 10 |
| | 8 | entire | 273 | 62 | 18 |
