# Peer review of "1. Introduction"

_Biogeosciences, 2020_

## Referee Comment (RC1) · Anonymous Referee #1 · 29 Sep 2020

Jacquet et al. present mesopelagic particulate organic carbon (POC) remineralisation fluxes in three different basins of the Mediterranean Sea using excess Barium (Baxs) as a proxy. This tracer, through the transfer function relating Baxs to oxygen consumption, has been successfully used in different regions of the World Ocean (Southern Ocean, North Atlantic, North Pacific) and has also been shown to be relevant in the Mediterranean Sea (Jacquet et al., in reviews). The study reveals interesting basinal variations in the magnitude of mesopelagic remineralisation (MR). Based on Baxs concentrations, greater 100-1000m MR fluxes are determined in the western basin

(Algerian basin) compared to the Tyrrhenian and Ionian basins. This greater deep remineralisation could be due to different processes such as 1) a strong convection in the western basin leading to a strong particle injection pump, or 2) dust deposition event in the eastern basins leading to a more efficient export of particles, thereby escaping mesopelagic remineralisation. Overall the data is interesting and necessary for better understanding the biological carbon pump. The two possible explanations of greater MR fluxes (lack of dust depositions, water mass convection) are also interesting, but are unfortunately not detailed enough and, even not mentioned in the conclusion. Moreover, the authors do not provide enough arguments and evidences to support their interpretations. For example, there is nothing demonstrating the good quality of the presented data, no data on dust deposition, and no explanations/data evidencing a winter deep convection in the western basin. My review consists in a relatively long list of questions, highlighting the lack of details in the manuscript. After revision, the new manuscript should provide all the details answering these questions. This will make the manuscript more convincing as for now the reader must believe your interpretation only with words and not with facts. Finally, the comparison of Baxs data from the same station but at different visits deserves more attention as there are no many studies (or none?) investigating the Baxs evolution over time. The significant difference of DWA Baxs and MR flux between visits however brings another question about the Baxs proxy: Is the seasonal time integration proposed in earlier studies correct?

Please, see my comments in the attached file.

Please also note the supplement to this comment:
https://bg.copernicus.org/preprints/bg-2020-271/bg-2020-271-RC1-supplement.pdf

**Supplement:**

Jacquet et al. present mesopelagic particulate organic carbon (POC) remineralisation fluxes in three different basins of the Mediterranean Sea using excess Barium ($Ba_{xs}$) as a proxy. This tracer, through the transfer function relating $Ba_{xs}$ to oxygen consumption, has been successfully used in different regions of the World Ocean (Southern Ocean, North Atlantic, North Pacific) and has also been shown to be relevant in the Mediterranean Sea (Jacquet et al., in reviews).

The study reveals interesting basinal variations in the magnitude of mesopelagic remineralisation (MR). Based on $Ba_{xs}$ concentrations, greater 100-1000m MR fluxes are determined in the western basin (Algerian basin) compared to the Tyrrhenian and Ionian basins. This greater deep remineralisation could be due to different processes such as 1) a strong convection in the western basin leading to a strong particle injection pump, or 2) dust deposition event in the eastern basins leading to more efficient export of particles, thereby escaping mesopelagic remineralisation.

Overall the data is interesting and necessary for better understanding the biological carbon pump. The two possible explanations of greater MR fluxes (lack of dust depositions, water mass convection) are also interesting, but are unfortunately not detailed enough and, even not mentioned in the conclusion. Moreover, the authors do not provide enough arguments and evidences to support their interpretations. For example, there is nothing demonstrating the good quality of the presented data, no data on dust deposition, and no explanations/data evidencing a winter deep convection in the western basin. My review consists in a relatively long list of questions, highlighting the lack of details in the manuscript. After revision, the new manuscript should provide all the details answering these questions. This will make the manuscript more convincing as for now the reader must believe your interpretation only with words and not with facts. Finally, the comparison of $Ba_{xs}$ data from the same station but at different visits deserves more attention as there are no many studies (or none?) investigating the $Ba_{xs}$ evolution over time. The significant difference of DWA $Ba_{xs}$ and MR flux between visits however brings another question about the $Ba_{xs}$ proxy: Is the seasonal time integration proposed in earlier studies correct?

**Major comments**

1) Abstract

This section is too short in my opinion and should mention the context of this study (dust deposition), why this study is important and what is its goal. Also the possible influence of dust deposition on remineralisation fluxes, which is one of your main interpretation should be mentioned. Finally, the authors should explain what the particle injection pump is and how this process can drive greater remineralisation fluxes in the Mediterranean Sea.

2) Material and methods

There should be more information about the sampling and the methods of both $Ba_{xs}$ and prokaryotic heterotrophic production (PHP) parameters. As such, the reader cannot fully understand the methods if he is not specialist. Moreover, there is no evidence you correctly determine these parameters: please indicate the precision and accuracy of your analyses.

Lines 76-89: Is there more information relevant to your study in the literature about these three basins, ie. Primary production, bloom timing, dust deposition events, POC export fluxes? If yes, please include them as they will help the reader to better understand the study area.

Lines 93-94: Indicate how much time separates both visits.

Line 97: Are these filters cleaned? If yes, how were they cleaned?

Lines 100-101: Please give the blanks, the precision and the accuracy of your analyses.

Lines 103-105: Show the equation to illustrate how you determine Baxs concentrations. Moreover, which Ba/Al ratio do you use? From the UCC, from aerosols? Discuss about the relevance of the used ratio for this study.

Line 106: How do you calculate this uncertainty? By error propagation, taking into account the analysis error of Ba and Al?

Section 2.3: This section is really not clear, you need to give many more details and to reorganise the paragraph (maybe by depth layer: 0-200m samples, 200-800m samples and >800m samples?). Moreover, if you have data at depths >800m, please show them on Fig. 2 and 3.
Line 119: Did you sample the same stations than for $Ba_{xs}$? How did you collect the samples, which sampling system did you use? How much volume did you collect?
Lines 121-123: Why are the sample from different depths incubated for different times? Why are the samples separated at these specific depths (ie., 0-200m; 200-800m and >800m)?
Line 124: 'Deep PHP' does it mean for samples from depth >800m?
Line 129: Is the protocol of Kirchman different from the one of Tamburini (line 125)? If yes, what is the difference? If no, combine both sentences at the end or beginning of the section.
Line 129: 'Epipelagic layers (0-250m)', why does it not correspond to the depth layer 0-200m described at the beginning of the section (line 122)?
Lines 131-134: With the information provided here, a reader who does not know about PHP analyses cannot understand how you estimate it. Please give more details: maybe an equation would help? Why do you mention isotope dilution here?

Section 2.4: Please mention you use DWA $Ba_{xs}$ to determine POC remineralisation fluxes and explain briefly what the DWA calculation is.

   3) Results

How do all these values compare to the literature?

Line 145: Why don't you show the PHP data at depths >800m?

Lines 154-155: What is a low lithogenic impact (give value please)? What does a >20% lithogenic fraction imply to your $Ba_{xs}$ estimations. This comes back to the explanations of using a correct lithogenic Ba/Al ratio for this study area (see one of my comments above)

Lines 167-171: Do the PHP peaks occur at the same depths than the $Ba_{xs}$ peaks (slightly above 100m)? Is there a link?

Lines 175-176: In both the 100-500 and 100-1000m depth layers?

Lines 177-179: The DWA $Ba_{xs}$ values do not remain stable if they decrease over time. Please make it clear.

   4) Discussion

You should give more details/more explanations supporting your interpretations.

Lines 185-187: How are Ca and Sr explaining the presence of ballasted phytoplankton-derived material? Do the Southern Ocean studies also report such high surface $Ba_{xs}$ concentrations?

Line 190: '..Ba$_{xs}$ presents the characteristic maximum..' Explain briefly what this characteristics maximum is. Someone who is not used to work on Ba$_{xs}$ concentrations cannot guess. For example, you could add: A typical profile of Ba$_{xs}$ shows a maximum in the mesopelagic layer (100-1000m) followed by a decrease of concentrations back to a background level, at deeper depths, usually below 1000m.

Lines 191-192: Please give average DWA Ba$_{xs}$ values for each basins and for both depth layers in order to directly compare the magnitudes.

Line 193-198: I suppose there a link between the dust event and the maximum Ba$_{xs}$ limited to the upper 500m in the TYR basin. Please find a transition/connection between both sentences. Moreover, compare average values of pAl concentrations (or %Ba$_{xs}$) between Stations 4, 5 and TYR and other stations to illustrate the differences between basins (cite Fig. 2 as well). Best would also to show a figure of dust events in the ~15 days before the cruise (maybe on Fig. 1?).

Line 202: How does the relationship between Ba$_{xs}$ and PHP reflect the temporal progression of POC remineralisation?

Line 204: Fig. 3 only shows the upper 1000m of PHP values.

Line 212: Please add a transition between both sentences. Maybe "We can however note that some data points, characterized by low DWA Ba$_{xs}$ values, do not follow the trend (from KEOPS2 and #3, #5 and #Tyrr2) '.

Lines 215-217: Can you demonstrate such temporal evolution? For example, satellite observations of surface Chl-a concentrations would show differences in bloom timing, which by taking into account a delay would suggest differences in remineralisation process.

Lines 279-219: The difference in DWA Ba$_{xs}$ content observed at the same station over time is very surprising to me. I thought Ba$_{xs}$ was a tracer integrating over a full season – how is it possible to observe such difference in only 2 days? Has this been discussed in earlier studies?

Line 229: 'small increase in MR rates at station #Tyrr between the two visits'. There is an increase from 32 to 114 mg C/m$^2$/d between both visits: this is a significant increase! Instead of focusing on this station, you could speak more generally, ie., averaging the all TYR basin.

Lines 227-233: Your hypothesis is convincing but how do you explain the restricted MR in the ION basin? Was there a dust deposition event there as well? A map of dust deposition averaging the ~15 days (or more?) before the sampling would give a good idea on how these basins were influenced by such events.

Line 237: Would you have an explanation for the low pAl concentrations in ALG basin while sampled just after a dust deposition?

Lines 238-242: This sentence is too long and not clear. Please re write.

Lines 243-252: This is one of your main conclusion and there is almost no explanation. You have to develop more: 1) is there a winter deep convection in the ALG basin (literature, data, how deep is the convection?)? Explain what the particle injection pump is and how this process can lead to greater MR rates?

5)  Conclusion

Why do you not mention the impact of dust deposition and winter deep convection as possible explanations of the greater MR fluxes observed in the ALG basin? This is, to me, the most interesting part of your manuscript.

6) Figures and Table

Line 282: You only show the upper 1000m of the PHP profiles. Also, change 'long' stations. You never use this description in the text.

Figure 1: Is it possible to add on the map where the dust deposition events occurred (surface colours maybe?), and where is the winter deep convection? Also what is the interest to show the T/S graph here if water masses are not discussed in the manuscript? What is the dashed line on the map?

Figure 3: Are the error bars shown?

Figure 4: The $R^2$ is very confusing. Does it take into account all data points presented on the figure or only those of KEOPS2? If it is the one of KEOPS, please update the $R^2$ by taking into account all data points shown here.

Figure 5: I am not sure it is necessary to show the MR fluxes of both depth layers below the figure, as they are indicated in Table 1.

Table 1: Can you please indicate the error in mg $C/m^2/d$ and not in %.

**Minor comments**

Line 22: Please keep the same appellation for the ALG basin throughout the manuscript: either Algerian or Algero-Provençal basin.

Line 70: Remove 'and' in 'and (3) to determine…'

Line 86: the abbreviation LSW is confusing with the Labrador Sea Water. Is there a way to distinguish both abbreviations?

Line 122: remove the 'n' in the end of depth '..below 800m depth..'

Line 124: Use the abbreviation PHP.

Line 127: Remove the second 'at' in 'were incubated at in situ temperature'

Line 129: Remove 'detailed' in 'The protocol is also detailed in Kirchman et al. (1993)'

Line 129: Epipelagic layers: 0-250m or 0-200m (as mentioned line 122)?

Line 140: Mention you investigate here the 100-500m and 100-1000m depth layers.

Line 149: 'Such high $Ba_{xs}$ contents..' instead of 'The very high $Ba_{xs}$ contents..'.

Line 150: What are the concentrations reached in the Southern Ocean?

Lines 152-153: This paragraph break in unnecessary.

Line 170: 'below 27'. Please give the exact value.

Line 172 and after: Why do you use the abbreviation DWAv instead of simply using DWA?

Line 206: Remove 'entire' in 'over the entire 100-500m depth interval'. This is confusing with Table 1.

Line 222: Add (100-500m) after '…for the upper .. ' and (100-1000m) after '..the entire..'

Line 222: Precise for what depth layer is the MR range.

Line 224:  'This is especially salient at station #9'. Please give values to illustrate your sentence.

Line 249: Write PHP in full and indicate the abbreviation

Line 298: '…layer for the Algero-Provençal (ALG), Tyrrhenian (TYR) and Ionian (ION) basins.'

---

## Referee Comment (RC2) · Anonymous Referee #2 · 11 Nov 2020

This manuscript presents a large data set of excess particulate Ba concentrations (Baxs) in the Mediterranean Sea showing spatial variations between basins. POC remineralization rates (MR) were estimated by Baxs inventories in mesopelagic waters and compared to data of prokaryotic heterotrophic production (PHP). This contribution is a good addition to the study of oceanic Ba cycle, in particular in marginal systems. However, I found that the data interpretation needs significant improvement and justification. Some explanations and statements are vague without solid evidence.

Major Issues:

Issue #1: Using Dehairs's transfer function. This is my biggest concern. I don't think this function can be directly used in the Mediterranean Sea without restriction. I also read the manuscript of Jacquet et al. (in review), which is also under review now at Biogeosciences. In Figure 2b of that manuscript, only a single data point from the Mediterranean Sea is located on the transfer line deduced from the Southern Ocean, while the Atlantic data point is clearly off the line. In addition, as shown in Figure 4 of this manuscript, the PEACETIME data set overall does not follow the trend of the Southern Ocean. In fact, Lemaitre et al. (2018) obtained a new transfer function for the Atlantic. Consequently, it is premature to make a statement of the universal validity of the Dehairs's transfer function. To fix this issue, I suggest the authors trying to develop a new transfer function specifically for the Mediterranean Sea using the large data set of this work, following what Lemaitre et al. (2018) did for the Atlantic scenario. A secondary option is keeping using Dehairs's transfer function, but the estimated POC MR needs very careful verification to prove such application is reasonable. This is exactly my second major concern.

Issue #2: Justification of the estimated POC MR. Whether the POC MR derived from the Baxs proxy is in order lacks justification. I suggest the authors comparing MR (Figure 5 and Table 2) with export production and/or primary production in the upper water column of the Mediterranean Sea. If these data are not available in the PEACETIME project, the authors can include literature data obtained from the Mediterranean Sea or from other similar systems for discussion.

Issue #3: Hypothesis of particle injection pump. To me this hypothesis, as the major implication of this study, was proposed without context in both the abstract (Lines 26-29) and the text (Lines 249-252). I didn't follow how Baxs variations between basins reflect the functioning of particle injection pump. I suggest the authors clarifying this point with more detailed discussion.

Minor Comments:

[Figure]

Lines 66-68: van Beek et al. (2009) also reported Baxs in the Mediterranean Sea.

Lines 152&158: in the "Results" section, expand description of the vertical distribution of particulate Al/Ca/Sr.

Lines 179-182: the description here is not consistent with data shown in Table 1, please double check.

Lines 196-198 & 227-229: what's the pattern of particulate Al and lithogenic Ba? Please be specific. "slight" means important or not important?

Lines 229-242: This part of discussion is unclear and needs reorganization. To me, the authors tended to explain two contrasting scenarios (increase and decrease in MR at two sites, respectively) using a same reason (i.e., dust input).

Line 256: what does "globally" mean?

Figures 2-3: I suggest the authors removing the data point of 2000 m to better show the Baxs maximum in the mesopelagic waters.

---

## Author Comment (AC1) · 30 Mar 2021

Title: Particulate biogenic barium tracer of mesopelagic carbon remineralization in the Mediterranean Sea (PEACETIME project)
Author(s): Stéphanie H. M. Jacquet et al.     MS No.: bg-2020-271 MS type: Research article
Special Issue: Atmospheric deposition in the low-nutrient-low-chlorophyll (LNLC) ocean: effects on marine life today and in the future (BG/ACP inter-journal SI)

Response to Referee #1

Jacquet et al. present mesopelagic particulate organic carbon (POC) remineralisation fluxes in three different basins of the Mediterranean Sea using excess Barium (Baxs) as a proxy. This tracer, through the transfer function relating Baxs to oxygen consumption, has been successfully used in different regions of the World Ocean (Southern Ocean, North Atlantic, North Pacific) and has also been shown to be relevant in the Mediterranean Sea (Jacquet et al., in reviews).
Reply: note that the paper Jacquet et al. #bg-2020-241 (On the barium-oxygen consumption relationship in the Mediterranean Sea: implications for mesopelagic marine snow remineralisation), is now published in Biogeosciences.

The study reveals interesting basinal variations in the magnitude of mesopelagic remineralisation (MR). Based on Baxs concentrations, greater 100-1000m MR fluxes are determined in the western basin (Algerian basin) compared to the Tyrrhenian and Ionian basins. This greater deep remineralisation could be due to different processes such as 1) a strong convection in the western basin leading to a strong particle injection pump, or 2) dust deposition event in the eastern basins leading to more efficient export of particles, thereby escaping mesopelagic remineralisation.
Overall the data is interesting and necessary for better understanding the biological carbon pump.
Reply: thanks. From a global climate perspective, the Baxs/MR tool will help to better balance the MedSea water column C budget. It will contribute to gain focus on the emerging picture of the C transfer efficiency (strength of the biological pump).

The two possible explanations of greater MR fluxes (lack of dust depositions, water mass convection) are also interesting, but are unfortunately not detailed enough and, even not mentioned in the conclusion. Moreover, the authors do not provide enough arguments and evidences to support their interpretations. For example, there is nothing demonstrating the good quality of the presented data, no data on dust deposition, and no explanations/data evidencing a winter deep convection in the western basin.
Reply: we agree that explanations are not detailed enough or supported by data. We'll add theses information to better constrain our conclusions.

My review consists in a relatively long list of questions, highlighting the lack of details in the manuscript. After revision, the new manuscript should provide all the details answering these questions. This will make the manuscript more convincing as for now the reader must believe your interpretation only with words and not with facts.
Reply: ok we'll answer each question

Finally, the comparison of Baxs data from the same station but at different visits deserves more attention as there are no many studies (or none?) investigating the Baxs evolution over time. The significant difference of DWA Baxs and MR flux between visits however brings another question about the Baxs proxy: Is the seasonal time integration proposed in earlier studies correct?
Reply: Baxs data in the medsea are scarce, but we'll consider them (especially the Dyfamed time series) to discuss Baxs evolution over time. We'll add discussion on seasonal time integration of the Ba signal.

**Major comments**

1) Abstract

This section is too short in my opinion and should mention the context of this study (dust deposition), why this study is important and what is its goal. Also the possible influence of dust deposition on remineralisation fluxes, which is one of your main interpretation should be mentioned. Finally, the authors should explain what the particle injection pump is and how this process can drive greater remineralisation fluxes in the Mediterranean Sea.

Reply: we completed abstract (and conclusion)

2) Material and methods

There should be more information about the sampling and the methods of both Baxs and prokaryotic heterotrophic production (PHP) parameters. As such, the reader cannot fully understand the methods if he is not specialist. Moreover, there is no evidence you correctly determine these parameters: please indicate the precision and accuracy of your analyses.

Reply: we added details on sampling and methods.

Lines 76-89: Is there more information relevant to your study in the literature about these three basins, ie. Primary production, bloom timing, dust deposition events, POC export fluxes? If yes, please include them as they will help the reader to better understand the study area.

Reply: we strengthened our discussion with data from literature.

Lines 93-94: Indicate how much time separates both visits.

Reply: done. three days separate both visits at station #Fast and two days at #Tyrr.

Line 97: Are these filters cleaned? If yes, how were they cleaned?

Reply: no, it is not necessary for Ba sampling to clean filters.

Lines 100-101: Please give the blanks, the precision and the accuracy of your analyses.

Reply: added. Both within +-5%

Lines 103-105: Show the equation to illustrate how you determine Baxs concentrations. Moreover, which Ba/Al ratio do you use? From the UCC, from aerosols? Discuss about the relevance of the used ratio for this study.

Reply: information added. From UCC Ba:Al molar ratio

Line 106: How do you calculate this uncertainty? By error propagation, taking into account the analysis error of Ba and Al?

Reply: by error propagation, yes including error on the Ba/Al ratio from UCC.

Section 2.3: This section is really not clear, you need to give many more details and to reorganise the paragraph (maybe by depth layer: 0-200m samples, 200-800m samples and >800m samples?). Moreover, if you have data at depths >800m, please show them on Fig. 2 and 3.

Line 119: Did you sample the same stations than for Baxs? How did you collect the samples, which sampling system did you use? How much volume did you collect?

Reply: Samples used for prokaryotes heterotrophic production were sampled with Niskin bottle into : the Algero-Provençal basin (later referred 80 to as ALG), the Tyrrhenian basin (TYR) and the Ionian basin (ION). Baxs and PHP have been collected in the same time on the same stations. From 0 until 200m-depth PHP has been measured in 1.5mL by using the microcentrifuge method with the 3H-leucine (3H-Leu) incorporation technique (Smith and Azam, 1992). For the depth > 200m, we used filtration method described in Tamburini et al. [2002] because below to 200m microcentrifuge method is less sensitive. Volume used is 20mL for the samples range between 200-800m-depth and 40mL for the samples up to 800m-depth.

Lines 121-123: Why are the sample from different depths incubated for different times? Why are the samples separated at these specific depths (ie., 0-200m; 200-800m and >800m)?
Reply: We have separated sample specific depth because incubation time is different according to the depth. The samples are separated because we used different methods to measure PHP.

Line 124: 'Deep PHP' does it mean for samples from depth >800m?
Reply: Yes

Line 129: Is the protocol of Kirchman different from the one of Tamburini (line 125)? If yes, what is the difference? If no, combine both sentences at the end or beginning of the section.
Reply: we modified it

Line 129: 'Epipelagic layers (0-250m)', why does it not correspond to the depth layer 0-200m described at the beginning of the section (line 122)?
Reply: we corrected it (0-200 m)

Lines 131-134: With the information provided here, a reader who does not know about PHP analyses cannot understand how you estimate it. Please give more details: maybe an equation would help? Why do you mention isotope dilution here?
Reply: We have worked at saturated concentration, so the initial concentration of natural leucine concentration is negligible.

Section 2.4: Please mention you use DWA Baxs to determine POC remineralisation fluxes and explain briefly what the DWA calculation is.
Reply: added. DWA corresponds to the Baxs inventory divided by the depth layer considered

3) Results

How do all these values compare to the literature?
Reply: added in results and discussion sections.

Line 145: Why don't you show the PHP data at depths >800m?
Reply: it is a question of scale in Fig2 (PHP gradients are too low below 500 m).

Lines 154-155: What is a low lithogenic impact (give value please)? What does a >20% lithogenic fraction imply to your Baxs estimations. This comes back to the explanations of using a correct lithogenic Ba/Al ratio for this study area (see one of my comments above)
Reply: added. We consider that Ba is mostly biogenic when Baxs >80%. When the lithogenic fraction increases >20-25%, this implies input of material from non-biogenic origin (non biological pump-generated).

Lines 167-171: Do the PHP peaks occur at the same depths than the Baxs peaks (slightly above 100m)? Is there a link?
Reply: no, as explained in discussion it is column integrated PHP gradients that could be compared to DWA Baxs (see section 4.2).

Lines 175-176: In both the 100-500 and 100-1000m depth layers?
Reply: on average yes

Lines 177-179: The DWA Baxs values do not remain stable if they decrease over time. Please make it clear.
Reply: we clarified it.

4) Discussion

You should give more details/more explanations supporting your interpretations.
Lines 185-187: How are Ca and Sr explaining the presence of ballasted phytoplankton-derived material? Do the Southern Ocean studies also report such high surface Baxs concentrations?
Reply: Ca and Sr can reflect the presence of coccolithophorids, forams or acantharians. We added explanation on Sr/Ca molar ratios. During peacetime Sr/Ca molar ratios range from 7 to 45 mmol mol$^{-1}$, which is in the range of ratios reported in organic material [Martin and Knauer, 1976].

Line 190: '..Baxs presents the characteristic maximum..' Explain briefly what this characteristics maximum is. Someone who is not used to work on Baxs concentrations cannot guess. For example, you could add: A typical profile of Baxs shows a maximum in the mesopelagic layer (100-1000m) followed by a decrease of concentrations back to a background level, at deeper depths, usually below 1000m.
Reply: ok added in results

Lines 191-192: Please give average DWA Baxs values for each basins and for both depth layers in order to directly compare the magnitudes.
Reply: done

Line 193-198: I suppose there a link between the dust event and the maximum Baxs limited to the upper 500m in the TYR basin. Please find a transition/connection between both sentences. Moreover, compare average values of pAl concentrations (or %Baxs) between Stations 4, 5 and TYR and other stations to illustrate the differences between basins (cite Fig. 2 as well). Best would also to show a figure of dust events in the ~15 days before the cruise (maybe on Fig. 1?).
Reply: we added lithogenic contribution (%). But we can not add a supplementary figure, this is presented in Guieu et al. (this issue).

Line 202: How does the relationship between Baxs and PHP reflect the temporal progression of POC remineralisation?
Reply: in terms of time laps between production, export, remineralization and subsequent barite formation in biogenic aggregates following the establishment of PHP gradients. Mesopelagic Ba$_{xs}$ content is smaller when most of the column integrated PHP is restricted to the upper mixed layer (indicating an efficient, close to complete remineralization within the surface), compared to situations where a significant part of integrated PHP was located deeper in the water column (reflecting significant deep bacterial activity and POC export).

Line 204: Fig. 3 only shows the upper 1000m of PHP values.
Reply: modified

Line 212: Please add a transition between both sentences. Maybe "We can however note that some data points, characterized by low DWA Baxs values, do not follow the trend (from KEOPS2 and #3, #5 and #Tyrr2) '.
Reply: ok done

Lines 215-217: Can you demonstrate such temporal evolution? For example, satellite observations of surface Chl-a concentrations would show differences in bloom timing, which by taking into account a delay would suggest differences in remineralisation process.
Reply: the time laps between the two sampling is too short. We completed discussion on integration time window.

Lines 279-219: The difference in DWA Baxs content observed at the same station over time is very

surprising to me. I thought Baxs was a tracer integrating over a full season – how is it possible to observe such difference in only 2 days? Has this been discussed in earlier studies?
Reply: during important (massive) export event in the SO we already observed such rapid Baxs signal evolution. We rephrased our discussion.

Line 229: 'small increase in MR rates at station #Tyrr between the two visits'. There is an increase from 32 to 114 mg C/m2/d between both visits: this is a significant increase! Instead of focusing on this station, you could speak more generally, ie., averaging the all TYR basin.
Reply: right! at station #Tyrr MR increases from 32 to 84 and 32 (upper) to 114 (entire). We modified our discussion

Lines 227-233: Your hypothesis is convincing but how do you explain the restricted MR in the ION basin? Was there a dust deposition event there as well? A map of dust deposition averaging the ~15 days (or more?) before the sampling would give a good idea on how these basins were influenced by such events.
Reply: the ION basin was not exposed to the dust event that occurs in the Tyrr basin 12 days before the sampling. It is indeed surprising to have similar restricted MR. We cautioned this in the discussion.

Line 237: Would you have an explanation for the low pAl concentrations in ALG basin while sampled just after a dust deposition?
Reply: it is a question of time laps (too short) between deposit and our water column sampling at station Fast. We mentioned it in discussion.

Lines 238-242: This sentence is too long and not clear. Please re write.
Reply: ok clarified

Lines 243-252: This is one of your main conclusion and there is almost no explanation. You have to develop more: 1) is there a winter deep convection in the ALG basin (literature, data, how deep is the convection?)? Explain what the particle injection pump is and how this process can lead to greater MR rates?
Reply: added.

5) Conclusion
Why do you not mention the impact of dust deposition and winter deep convection as possible explanations of the greater MR fluxes observed in the ALG basin? This is, to me, the most interesting part of your manuscript.
Reply: added. The conclusion was indeed incomplete.

6) Figures and Table

Line 282: You only show the upper 1000m of the PHP profiles. Also, change 'long' stations. You never use this description in the text.
Reply: ok modified

Figure 1: Is it possible to add on the map where the dust deposition events occurred (surface colours maybe?), and where is the winter deep convection? Also what is the interest to show the T/S graph here if water masses are not discussed in the manuscript? What is the dashed line on the map?
Reply: We don't think it is crucial to add a map of dust deposition (see the paper of Guieu et al. for that) or to add it on the actual map. A T/S diagram always helps in visualizing the hydrographical situation of the studied zone. We indeed only report water mass name and major characteristics. We completed the description. The dashed line represents transects reported in Fig2. We added it in the caption.

Figure 3: Are the error bars shown?
Reply: yes they are (around 5%)

Figure 4: The R2 is very confusing. Does it take into account all data points presented on the figure or only those of KEOPS2? If it is the one of KEOPS, please update the R2 by taking into account all data points shown here.
Reply: We added in the text and Fig4 that the correlation is given for KEOPS2. The aim is not to define a single correlation combining the different cruises but to compare different seasonal situations in a same sector and understand potential evolution. That is why it is non-sense to generalise a single correlation.

Figure 5: I am not sure it is necessary to show the MR fluxes of both depth layers below the figure, as they are indicated in Table 1.
Reply: we think it is important to see the contrast at stations where MR occurs deeper (higher 100-1000 m), and conversely, where MR is mainly restricted in the upper 500 m.

Table 1: Can you please indicate the error in mg C/m2/d and not in %.
Reply: we prefer to keep it in %, it is easier to compare fluxes.

**Minor comments**

Line 22: Please keep the same appellation for the ALG basin throughout the manuscript: either Algerian or Algero-Provençal basin.
Reply: done. The algero-Provençal basin is referred to as ALG all along the ms.

Line 70: Remove 'and' in 'and (3) to determine…'
Reply: done

Line 86: the abbreviation LSW is confusing with the Labrador Sea Water. Is there a way to distinguish both abbreviations?
Reply: this is the specific term ('nomenclature') used in literature for the Mediterranean system. By the way, no Labrador sea water in the MedSea.

Line 122: remove the 'n' in the end of depth '..below 800m depth..'
Reply: done

Line 124: Use the abbreviation PHP.
Reply: done

Line 127: Remove the second 'at' in 'were incubated at in situ temperature'
Reply: done

Line 129: Remove 'detailed' in 'The protocol is also detailed in Kirchman et al. (1993)'
Reply: done

Line 129: Epipelagic layers: 0-250m or 0-200m (as mentioned line 122)?
Reply: 0-200m; we corrected it.

Line 140: Mention you investigate here the 100-500m and 100-1000m depth layers.
Reply: done

Line 149: 'Such high Baxs contents..' instead of 'The very high Baxs contents..'.
Reply: done

Line 150: What are the concentrations reached in the Southern Ocean?
Reply: up to 9000 p M (Dehairs et al., 1997; Atlantic sector SO) and up to 6000 pM (Jacquet et al., 2008; Indian SO). We added it in the ms.

Lines 152-153: This paragraph break in unnecessary.
Reply: ok

Line 170: 'below 27'. Please give the exact value.
Reply: added. 25 mg C L_1

Line 172 and after: Why do you use the abbreviation DWAv instead of simply using DWA?
Reply: we modified it (v= values)

L_i_n_e_ _2_0_6_:_ _R_e_m_o_v_e_ _'e_n_t_i_r_e_' _i_n_ _'o_v_e_r_ _t_h_e_ _e_n_t_i_r_e_ _1_0_0_-_5_0_0_m_ _d_e_p_t_h_ _i_n_t_e_r_v_a_l_'._ _T_h_i_s_ _i_s_ _c_o_n_f_u_s_i_n_g_ _w_i_t_h_ _T_a_b_l_e_ _1_._ _
Reply: done
L_i_n_e_ _2_2_2_:_ _A_d_d_ _(_1_0_0_-_5_0_0_m_)_ _a_f_t_e_r_ _'…f_o_r_ _t_h_e_ _u_p_p_e_r_ _._._ _ '_a_n_d_ _(_1_0_0_-_1_0_0_0_m_)_ _a_f_t_e_r_ _'._._t_h_e_ _e_n_t_i_r_e_._._'_
Reply: done

L_i_n_e_ _2_2_2_:_ _P_r_e_c_i_s_e_ _f_o_r_ _w_h_a_t_ _d_e_p_t_h_ _l_a_y_e_r_ _i_s_ _t_h_e_ _M_R_ _r_a_n_g_e_._
Reply: it is a global range

L_i_n_e_ _2_2_4_:_ _'T_h_i_s_ _i_s_ _e_s_p_e_c_i_a_l_l_y_ _s_a_l_i_e_n_t_ _a_t_ _s_t_a_t_i_o_n_ _#_9_'._ _P_l_e_a_s_e_ _g_i_v_e_ _v_a_l_u_e_s_ _t_o_ _i_l_l_u_s_t_r_a_t_e_ _y_o_u_r_ _s_e_n_t_e_n_c_e_._ _
Reply: done

L_i_n_e_ _2_4_9_:_ _W_r_i_t_e_ _P_H_P_ _i_n_ _f_u_l_l_ _a_n_d_ _i_n_d_i_c_a_t_e_ _t_h_e_ _a_b_b_r_e_v_i_a_t_i_o_n_ _
Reply: not necessary to mention it again in conclusion

L_i_n_e_ _2_9_8_:_ _'…l_a_y_e_r_ _f_o_r_ _t_h_e_ _A_l_g_e_r_o_-_P_r_o_v_e_n_ça_l_ _(_A_L_G_)_,_ _T_y_r_r_h_e_n_i_a_n_ _(_T_Y_R_)_ _a_n_d_ _I_o_n_i_a_n_ _(_I_O_N_)_ _b_a_s_i_n_s_._'_
Reply: already mentioned in the text.

---

## Author Comment (AC2) · 30 Mar 2021

Title: Particulate biogenic barium tracer of mesopelagic carbon remineralization in the Mediterranean Sea (PEACETIME project)
Author(s): Stéphanie H. M. Jacquet et al.     MS No.: bg-2020-271 MS type: Research article
Special Issue: Atmospheric deposition in the low-nutrient-low-chlorophyll (LNLC) ocean: effects on marine life today and in the future (BG/ACP inter-journal SI)

Response to Referee #2
This manuscript presents a large data set of excess particulate Ba concentrations (Baxs) in the Mediterranean Sea showing spatial variations between basins. POC remineralization rates (MR) were estimated by Baxs inventories in mesopelagic waters and compared to data of prokaryotic heterotrophic production (PHP). This contribution is a good addition to the study of oceanic Ba cycle, in particular in marginal systems.
Reply: thanks, but do you mean "marginal" in term of coastal/marge/plateau zones, impacted by convection and lithogenic /continental inputs? Or "marginal" in term of secondary type/minor system? Because the Mediterranean Sea is not marginal system. Due to limited exchanges with adjacent basins and the existence of an intense overturning circulation qualitatively resembling the global one but characterized by shorter time-scales, the Mediterranean Sea is to date considered as a laboratory to observe and understand the impact of transient climate variability on ecosystems and biogeochemical cycles. In a context of climate changes, better balancing the regional carbon budget and C storage capacity is of crucial importance in the MedSea.

However, I found that the data interpretation needs significant improvement and justification. Some explanations and statements are vague without solid evidence.
Reply: ok we'll strengthen our explanations.

Major Issues:

Issue #1: Using Dehairs's transfer function. This is my biggest concern. I don't think this function can be directly used in the Mediterranean Sea without restriction. I also read the manuscript of Jacquet et al. (in review), which is also under review now at Biogeosciences. In Figure 2b of that manuscript, only a single data point from the Mediterranean Sea is located on the transfer line deduced from the Southern Ocean, while the Atlantic data point is clearly off the line. In addition, as shown in Figure 4 of this manuscript, the PEACETIME data set overall does not follow the trend of the Southern Ocean.
In fact, Lemaitre et al. (2018) obtained a new transfer function for the Atlantic. Consequently, it is premature to make a statement of the universal validity of the Dehairs's transfer function. To fix this issue, I suggest the authors trying to develop a new transfer function specifically for the Mediterranean Sea using the large data set of this work, following what Lemaitre et al. (2018) did for the Atlantic scenario. A secondary option is keeping using Dehairs's transfer function, but the estimated POC MR needs very careful verification to prove such application is reasonable. This is exactly my second major concern.
Reply: the ms. Jacquet et al. #bg-2020-241 is now published in Biogeosciences. We show that the Dehairs function can indeed be used in the MedSea. Furthermore, in Lemaitre et al. (2008) authors clearly mention that the transfer function they obtained (from apparent oxygen utilisation divided by the water mass age) in the north Atlantic is not significantly different to that reported by Dehairs from the Southern Ocean. There is no need to develop a new function here because by testing the Ba vs JO2 relationship at ANTARES we obtained very close trend. Also, time integration of AOU is less precise than direct measurements. Lemaitre was furthermore one of the reviewers of the #bg-2020-241 paper and approved our conclusion that the function is relevant in the MedSea.
We agree that efforts should be put in the ms on confronting our MR fluxes with more data and literature.

Issue #2: Justification of the estimated POC MR. Whether the POC MR derived from the Baxs proxy is in order lacks justification. I suggest the authors comparing MR (Figure 5 and Table 2) with export production and/or primary production in the upper water column of the Mediterranean Sea. If these data are not available in the PEACETIME project, the authors can include literature data obtained from the Mediterranean Sea or from other similar systems for discussion.

Reply: indeed production, export or other surface C fluxes were not measured during the peacetime cruise. It was not in the core of the project. We'll give some range of values found in literature.

Issue #3: Hypothesis of particle injection pump. To me this hypothesis, as the major implication of this study, was proposed without context in both the abstract (Lines 26- 29) and the text (Lines 249-252). I didn't follow how Baxs variations between basins reflect the functioning of particle injection pump. I suggest the authors clarifying this point with more detailed discussion.

Reply: ok, we'll better explain how particle injection could impact Baxs variations between basins. Briefly, this process controls the depth where remineralization of POC occurs (and subsequent barite formation). The injection process (strong convection) has been reported in literature to be particularly salient in the western basin. In a previous paper we also reported its potential impact on the dissolved Ba distribution. As the Baxs vertical distribution clearly reflects a deeper export of material in the western basin during peacetime, we formulated that the origin of this material could be the particle injection pump (and subsequent remineralization, barite formation, etc…). We'll better discuss it in the ms.

Minor Comments:
Lines 66-68: van Beek et al. (2009) also reported Baxs in the Mediterranean Sea.

Reply: added. Note that it is in part the same data set as published in Sternberg et al., 2008- but discussed in the light of Ra/Ba ratios.

Lines 152&158: in the "Results" section, expand description of the vertical distribution of particulate Al/Ca/Sr.

Reply: added in "results" and "discussion 4.1" sections.

Lines 179-182: the description here is not consistent with data shown in Table 1, please double check.

Reply: ok modified

Lines 196-198 & 227-229: what's the pattern of particulate Al and lithogenic Ba? Please be specific. "slight" means important or not important?

Reply: added. A low (to very low) lithogenic contribution does not exceed 20%.

Lines 229-242: This part of discussion is unclear and needs reorganization. To me, the authors tended to explain two contrasting scenarios (increase and decrease in MR at two sites, respectively) using a same reason (i.e., dust input).

Reply: the decreased (lower intensity) and upper-mesopelagic restricted layer MR is potentially due to the dust input at station #Tyr. The impact is not supported by Al data at Fast station. We modified and clarified the discussion.

Line 256: what does "globally" mean?

Reply: nothing ☺ – we removed it

Figures 2-3: I suggest the authors removing the data point of 2000 m to better show the Baxs maximum in the mesopelagic waters.

Reply: it is important to keep it for data presentation and to see how concentrations decrease to the background. The mesopelagic maximum is enough salient in Fig2&3.

---

## Editor Decision (ED1)

[revised manuscript text omitted]

The present work is part of the PEACETIME project (ProcEss studies at the Air-sEa Interface after dust deposition in the MEditerranean sea) (http://peacetime-project.org/). PEACETIME aimed at studying the impact of atmospheric Saharan dust on the Mediterranean biogeochemistry [Guieu et al., 2020a]. Dust deposition is a major source of macro and micro nutrients and ballasted material to surface waters that likely impacts the biological carbon pump through organic matter production (i.e. primary production) and its subsequent export and remineralization in the water column [Pabortsava et al., 2017; Gazeau et al., 2021]. Overall, the aims of the present contribution to the PEACTIME project were: (1) to document particulate biogenic $Ba_{xs}$ in different ecoregions of the western and central parts of the Mediterranean Sea. Previous $Ba_{xs}$ data in the Mediterranean Sea are relatively scarce with limited vertical sampling resolution [Sanchez-Vidal et al., 2005] or restricted locations [Dehairs et al., 1987; Sternberg et al., 2007, 2008; van Beek et al., 2009]; (2) to determine the relationship between $Ba_{xs}$ and environmental variability, including dust deposition, (3) to estimate $Ba_{xs}$-based POC remineralization rates (MR) at mesopelagic depths according to the Dehairs' transfer function [Dehairs et al., 1997] we have recently validated for the Mediterranean Sea [Jacquet et al., 2021], and (4) assess potential differences in remineralization length scale of POC in the various ecoregions of the Mediterranean Sea.

**2. Material and methods**

**2.1 Study area**

The PEACETIME cruise (https://doi.org/10.17600/17000300) was conducted during late spring conditions from May 10 to June 11, 2017 (French R/V Pourquoi pas?) in the western and central Mediterranean (Figure 1a). The Mediterranean Sea is a semi-landlocked sea with limited, but crucial, exchange with the Atlantic Ocean, two deep overturning cells, one shallow circulation and a complex upper layer circulation with several permanent and quasi-permanent eddies. The Mediterranean Sea is furthermore characterized by contrasting ecosystems, from strongly oligotrophic deep interiors to eutrophic Adriatic [Durrieu de madron et al., 2011; Reygondeau et al., 2017]. The studied zone crossed the typical eastward increasing oligotrophic trend as reported in previous studies [Moutin and Raimbault, 2002; Pujo-Pay et al., 2011; Tanhua et al., 2013a; Guieu et al., 2020a]. However, this trend was not homogeneous, as for instance in the Ionian Sea (a crossroad of waters of contrasted biological history) where blooming and non-blooming mosaic areas co-occur in spring [Berline et al., 2021].

[revised manuscript text omitted]

VanWambeke et al. [2021]. Figure 4 shows vs. DWA $Ba_{xs}$ calculated over the 100-500 m depth data  in the Southern Ocean [Jacquet et al., 2008a, 2015] and recently in the northeast Atlantic and northwestern Mediterranean Sea (PAP and ANTARES

observatory sites, respectively) [Jacquet et al., 2021].

Results during PEACETIME follow higher DWA $Ba_{xs}$ in situation where a significant part of column-integrated PHP is located deeper in the water column (high Int.PHP/IntPHP ratio, Figure 4).

During KEOPS2, stations located in a meander and reflected  different (earlier) stages of bloom  Figure 4).

Similarly, station #5 and #Tyrr2  reflect evolution of mesopelagic remineralization processes in the TYR basin compared to the other basins. Measurements  during the second visit at station #Tyrr4 two days later corroborate this hypothesis  an increase  remineralization  in the upper mesopelagic layer (DWV $Ba_{xs}$ increased from 284 to 542 pM). At the DYFAMED

station, Sternberg et al. [2008] reported the seasonal evolution of $Ba_{xs}$ profiles on a monthly basis between February and June 2003.  mesopelagic $Ba_{xs}$

builds up (and barite stock increase) following the spring phytoplankton bloom development, POC fluxes and subsequent remineralization . DWA $Ba_{xs}$ reported in the present study are  higher than those  reported in

Sternberg et al. [2008]. They  variability over the two days period at station

#Tyrr of the same order of magnitude as  at DYFAMED.

$_{xs}$

in different sectors of the Southern Ocean.  the column-integrated PHP vs. DWA $Ba_{xs}$ ratio station #Tyrr confirms that the second  higher remineralization in the upper mesopelagic layer than during the first occupation.

**4.3 Mesopelagic C remineralization**

$_{xs}$ POC remineralization rates (MR) using Eq. (1).

for the upper (100-500 m) and entire (100-1000 m) mesopelagic layer. $^{-2}$$^{-1}$

$^{-2}$$^{-1}$

large difference in MR  in the ALG basin . This is at station #9 with MR of 91 mg C m$^{-2}$ d$^{-1}$ in the upper  layer and

306 mg C m$^{-2}$ d$^{-1}$  the entire mesopelagic layer (Figure 6). Results  significant remineralization occurred between 500 and 1000 m in the ALG basin In contrast,  the ION

and TYR basins  remineralization  mainly  in the upper

. Similar conclusion was  Jullion et al. [2017]

dissolved Ba and Parametric Optimum Multiparameter (POMP)-derived POC

remineralization  along a zonal transect between the Lebanon coast and Gibraltar (from to 348 mg C m$^{-2}$ d$^{-1}$; M84/3 cruise, April 2011). Independent of any dust consideration,

 showed significant differences in the mesopelagic MR between

 Mediterranean  an additional export pathway

 in the western basin. The western basin is indeed the site of  shelf and open ocean  convection, transferring

 to  [Durrieu de Madron et al.,

2013; Stabholz et al., 2013].

 the larger MR fluxes  in the ALG basin during PEACETIME are in line with an ecoregion  recurrent  of material  by convection  remineralization . In contrast in the

TYR basin remineralization  mainly located in the upper mesopelagic layer. Stations in the TYR basin  particulate Al concentrations and estimated lithogenic Ba fraction reflect the impact of this event. At station #Tyrr the DAW

Ba$_{xs}$ vs. column-integrated PHP  between the two visit  MR rates mainly localized in the upper 500 m. Despite this increase, we can wonder whether the overall impact of the dust event would result in global lower (intensity) and upper mesopelagic layerrestricted MR processes. This would suggest that by providing ballast material (dust), and thereby decreasing of the exposition time of particles to prokaryotic remineralization

[Pabortsava et al., 2017], the dust event reduced MR at station #Tyr. Another occurred on June 5, a few hours after the first sampling at station #Fast in the ALG basin.

However, station #Fast does not present any evidence of an impact at mesopelagic depths on particulate Al concentrations and estimated lithogenic Ba . In contrast to  in the surface layer,  mesopelagic remineralization and subsequent $Ba_{xs}$ formation) to a single dust event would require more time.  stations  not reflect the impact of any dust event  and are not subject to potential deep convection (as reported in the ALG basin), DWA $Ba_{xs}$ and MR  restricted to the upper mesopelagic layer.  Berline et al. [this issue]  report  small-scale heterogeneity of particles abundance at ION stations, emphasizing the spatial decoupling between particle production and particle distribution and adding complexity in  time lag between production and export of particles, and thus C transfer  [Stange et al., 2017; Henson et al., 2011].  significant surface production event  surface particles at station #8  to a past production event  exported . 
[revised manuscript text omitted]

[Figure]

[Figure]

Figure 2

[Figure]

Figure 3

[Figure]

Figure 4

[Figure]

Figure 5

[Figure]

| | 1 | 2 | 3 | 10 | Fast1 | Fast3 | 9 | 4 | 5 | Tyrr2 | Tyrr4 | 6 | 8 | 7 | Ion1 |
|---|---|---|---|---|---|---|---|---|---|---|---|---|---|---|---|
| ■ 100-1000m | 117 | 245 | 104 | 129 | 183 | 117 | 306 | 71 | 45 | 32 | 114 | 85 | 62 | 167 | 52 |
| ■ 100-500m | 94 | 131 | 58 | 60 | 105 | 99 | 91 | 25 | 32 | 32 | 84 | 58 | 41 | 139 | 58 |

Table 1

| Basin | Station # | Mesopelagic layer | DWAv Ba$_{xs}$ [pM] | MR [mg C m$^{-2}$ d$^{-1}$] | Stnd error (%) |
|---|---|---|---|---|---|
| Algero-Provençal | 1 | upper | 542 | 94 | 20 |
| | 1 | entire | 374 | 117 | 27 |
| | 2 | upper | 717 | 131 | 26 |
| | 2 | entire | 645 | 245 | 28 |
| | 3 | upper | 402 | 58 | 13 |
| | 3 | entire | 353 | 104 | 25 |
| | 4 | upper | 243 | 25 | 8 |
| | 4 | entire | 281 | 71 | 20 |
| | 9 | upper | 981 | 91 | 19 |
| | 9 | entire | 979 | 306 | 23 |
| | Fast1 | upper | 638 | 105 | 21 |
| | Fast1 | entire | 527 | 183 | 38 |
| | Fast3 | upper | 596 | 99 | 20 |
| | Fast3 | entire | 381 | 117 | 27 |
| | 10 | upper | 418 | 60 | 13 |
| | 10 | entire | 410 | 129 | 29 |
| Tyrrhenian | 5 | upper | 283 | 32 | 9 |
| | 5 | entire | 226 | 45 | 17 |
| | Tyrr2 | upper | 284 | 32 | 9 |
| | Tyrr2 | entire | 200 | 32 | 15 |
| | Tyrr4 | upper | 542 | 84 | 17 |
| | Tyrr4 | entire | 380 | 114 | 26 |
| | 6 | upper | 404 | 58 | 13 |
| | 6 | entire | 313 | 85 | 22 |
| Ionian | 7 | upper | 769 | 139 | 28 |
| | 7 | entire | 485 | 167 | 36 |
| | ION | upper | 456 | 58 | 13 |
| | ION | entire | 315 | 52 | 13 |
| | 8 | upper | 363 | 41 | 10 |
| | 8 | entire | 273 | 62 | 18 |

---

## Author Response (AR2)

**Associate Editor Decision: Publish subject to minor revisions (review by editor)** (07 Jun 2021) by Christine Klaas Comments to the Author (pdf):bg-2020-271-comments-to-author.pdf

Comments to the Author:Dear authors, the revised version of the manuscript appropriately addressed comments from both reviewer with perhaps one exception:

**Response to Editor**

Section 2.1: this section describes the study area in general terms (including a detailed description of waters masses). However, there is little information from and of relevance to the study here (i.e. the relevant info is revealed only piecewise in the discussion). It would help the reader if the available relevant information (also from other PEACETIME manuscript in the special issue) be reported/summarised here. This concerns in particular:1) Primary productivity and fluxes or particle load at the different stations before and during PEACETIME. 2) Significant atmospheric inputs and relevant elements at the different stations before and during PEACETIME.

Reply: we added further information related to PP and atmospheric inputs in session 2.1 which are relevant for discussion. We also referred to the works of van Wambecke and Bressac for further details. Session 2.1 has been revised.

And last but not least, for the discussion, how is the impact of deep winter convection in the western basin: were, when and how deep? Where does the POM come from: resuspended coastal sediments? Which area does this transport influence? Is there any info on particles in this layer? Finally, how does DOM affect the relation between Baxs and PHP in this particular study (i.e. DOM might not affect Baxs, but it would affect PHP).

Reply: the impact of deep winter convection on POM (and thus remineralization and subsequent Ba formation) is a conclusion/deduction from the present results and from previous works on Ba (Jacquet et al., 2016; Jullion et al., 2017) and from the fact that the western basin is a well known site of deep shelf and open ocean convection, transferring organic matter to deeper layers [Durrieu de Madron et al., 2013; Stabholz et al., 2013]. But we have no direct info on particles in this layer during the Peacetime cruise. We only dispose of upper POC fluxes that were similar at the 3 main stations at 200 m depth.

Minor comments and request for clarifications and some suggestion for improvement of the text are given in the annotated manuscript. I would urge the authors to use the same tense throughout (use either past to present tense to describe PEACETIME results) as well as the same abbreviations in text as in figures (see annotated manuscript).

Reply: thank you for all the comments and corrections made in the manuscript. We integrated them in our revised version. The same tense (past) is now used and we checked all abbreviations.

---

## Editor Decision (ED3)

**ABSTRACT**

We report on the sub-basins variability of particulate organic carbon (POC) remineralization in the western and central Mediterranean Sea during  late spring  PEACETIME cruise. POC remineralization rates were estimated using the excess biogenic particulate barium ($Ba_{xs}$) inventories in mesopelagic  (100-1000 m) and compared with prokaryotic heterotrophic production (PHP). $Ba_{xs}$-based mesopelagic remineralization rates (MR)  from $25 \pm 2$ to $306 \pm 70$ mg C m$^{-2}$ d$^{-1}$. MR  larger in the  (ALG) basin  the Tyrrhenian (TYR) and Ionian (ION) basins. Our $Ba_{xs}$ inventories and integrated PHP data also  that significant mesopelagic remineralization  down to 1000 m depth in the ALG basin in contrast to the ION and TYR basins where remineralization  mainly located  500 m . We proposed that the  and deeper MR rates in the ALG basin  sustained by an additional particles export event driven by deep convection. The TYR basin (in contrast to the ALG and ION basins)  the impact of a previous dust event as reflected by our particulate Al water column concentrations. The ION and TYR basins  small-scale heterogeneity   remineralization processes,  reflected by our $Ba_{xs}$ inventories and integrated PHP data at the #Tyrr long duration station. This heterogeneity  linked to the mosaic of blooming and non-blooming patches reported in this area during the cruise.  to the western Mediterranean Sea (ALG basin), the central Mediterranean Sea (ION and TYR basins)  lower  remineralization  during the late spring PEACETIME cruise.